**Fracturing and crystal plastic behaviour of garnet under seismic stress in the**
**dry lower continental crust (Musgrave Ranges, Central Australia)**
Friedrich Hawemann[1*], Neil Mancktelow[1], Sebastian Wex[1], Giorgio Pennacchioni[2], Alfredo
Camacho[3]
1) Department of Earth Sciences, ETH Zurich, CH8092 Zurich, Switzerland
2) Department of Geosciences, University of Padova, Padova, Italy
3) Department of Geological Sciences, University of Manitoba, Winnipeg, Manitoba, R3T
2N2, Canada
* corresponding author friedrich.hawemann@erdw.ethz.ch

**Highlights**

• garnet deformed by fracturing and crystal-plasticity under dry lower crustal conditions
• Ca-diffusion profiles indicate multiple generations of fracturing
• diffusion is promoted along zones of higher dislocation density
• fracturing indicates transient high-stress (seismic) events in the lower continental
crust
**Abstract**
Garnet is a high strength mineral compared to other common minerals such as quartz and
feldspar in the felsic crust. In felsic mylonites, garnet typically occurs as porphyroclasts that

mostly evade crystal-plastic deformation, except under relatively high temperature conditions. The microstructure of granulite facies garnet in felsic lower-crustal rocks of the Musgrave Ranges (Central Australia) records both fracturing and crystal-plastic deformation. Granulite facies metamorphism at ~ 1200 Ma generally dehydrated the rocks and produced mm-sized garnets in peraluminous gneisses. A later ~ 550 Ma overprint under sub-eclogitic conditions (600-700 °C, 1.1-1.3 GPa) developed mylonitic shear zones and abundant pseudotachylyte, coeval with the neocrystallization of fine-grained, high-calcium garnet. In the mylonites, granulite-facies garnet porphyroclasts are enriched in calcium along rims and fractures. However, these rims are locally narrower than otherwise comparable rims along original grain boundaries, indicating contemporaneous diffusion and fracturing of garnet. The fractured garnets exhibit internal crystal-plastic deformation, which coincides with areas of enhanced diffusion, usually along zones of crystal lattice distortion and dislocation walls associated with subgrain rotation recrystallization. Fracturing of garnet under dry lower crustal conditions, in an otherwise viscously flowing matrix, requires transient high differential stress, most likely related to seismic rupture, consistent with the coeval development of abundant pseudotachylyte.

**Keywords**

Garnet, Fracture, Crystal-Plasticity, Dry Lower Continental Crust, Pseudotachylyte, Seismicity

## 1 Introduction

A fundamental problem in geology is the limited preservation of processes in the rock record. This is especially the case for transient events, like earthquakes, traces of which are hardly

preserved due to later reworking. The best indicators for seismicity in the rock record are
pseudotachylytes (Sibson, 1975; Toy et al., 2011), although not every seismic event produces
frictional melts and, once formed, ductile creep or later brittle fracturing may erase most
traces (Sibson and Toy, 2006; Kirkpatrick and Rowe, 2013).
Garnet is stable in many metamorphic rocks over a large part of the pressure-temperature
space, is commonly preserved, and is suitable for a range of geothermobarometers and
geochronometers and their combination for geospeedometry (Lasaga, 1983; Caddick et al.,
2010; Baxter and Scherer, 2013). Being a high strength mineral (Karato et al., 1995; Wang and
Ji, 1999), both brittle and crystal plastic deformation are rarely observed in garnet when
compared to the common matrix minerals of the crust, such as quartz and feldspar. However,
Dalziel and Bailey (1968) already interpreted elongate garnets in high grade mylonites to be
the result of crystal plastic behaviour and advancements since then in electron microscopy,
and especially EBSD (electron backscatter diffraction), have allowed detailed investigation of
garnet textures (Kunze et al., 1993; Prior et al., 2000, 2002).
Experimental deformation of garnet indicates that differential stresses on the order of a few
GPa are required to produce shear fractures, and that the onset of crystal plastic behaviour
for strain rates typical of actively deforming regions ($10^{-12} - 10^{-15}$ s$^{-1}$; e.g. Behr and Platt, 2011)
should only occur at corresponding temperatures above ca. 750-640 °C (Karato et al., 1995;
Wang and Ji, 1999). The observation of fractured garnets in natural samples may therefore
be linked to seismic stresses, as suggested by Austrheim et al. (1996), who described
fracturing of garnets during pseudotachylyte formation and fluid-assisted eclogitization of
granulites. Trepmann and Stöckhert (2002) also interpreted the microstructure of fractured
and offset garnets as evidence for syn-seismic loading and post-seismic creep. More recently,
both brittle (Austrheim et al., 2017; Engi et al., 2017; Angiboust etal., 2017; Giuntoli et al.,
2018; Hawemann et al., 2018; Petley-Ragan et al., 2019) and associated crystal-plastic
behaviour (Austrheim etal., 2017; Petley-Ragan et al., 2019) of garnet has been related to
seismic events in lower continental crust or deeply subducted continental fragments. Papa et
al. (2018) interpreted similar deep-seated dilatant fracturing of garnet immediately adjacent
to pseudotachylyte to be related to thermal shock due to frictional heating rather than to
damage associated with propagation of the seismic rupture. Konrad-Schmolke et al. (2007)
described enhanced diffusion of Mg along subgrain boundaries in garnet (but not of slow
diffusing elements, such as Ca, Ti and Y) from high pressure meta-granitoids of the deeply
subducted Sesia Zone (Western Alps). However, in contrast to more recent studies in the Sesia
Zone, which propose that precursor fracturing was crucial for dissolution–precipitation and
diffusion processes in garnet (Engi et al., 2018; Giuntoli et al., 2018), they considered that
there were no signs of crystal-plastic deformation in their garnet samples and concluded that
a diffusion-induced dislocation migration and/or diffusion-induced recrystallisation process
was responsible for development of the observed subgrain texture.
Garnets can retain their microstructure and chemical composition during retrograde
deformation and metamorphism and can therefore preserve indicators for seismic events,
which are otherwise possibly erased from the rock record.Here we present a study of garnet
microstructures from lower crustal rocks of the Musgrave Block in Australia, which:

86       (1) illustrates the close association between brittle and crystal-plastic deformation of

87           garnet under well-established pressure-temperature conditions;

88       (2) infers deformation mechanisms from the observed microstructure;

89       (3) explores the close link between deformation and diffusion in garnet;

(4) complements other independent observations indicating transient high stresses in the

91        lower crust.


## 2  Geological setting

2.1 Regional geology
The Musgrave Block is located in an intraplate position close to the centre of the Australian
continent (inset Fig. 1). Amalgamation of the different cratonic blocks took place during the
Musgravian Orogeny (1120-1200 Ma), which pervasively overprinted ca. 1550 Ma gneisses
(Gray, 1978; Camacho and Fanning, 1995). The Petermann Orogeny (~550 Ma) produced a
series of crustal-scale fault zones, most prominently the Woodroffe Thrust and the Mann
Fault (Collerson et al., 1972; Major, 1973; Bell, 1978; Camacho and Fanning, 1995; Raimondo
et al., 2010; Hawemann et al., 2018, 2019; Wex et al., 2017, 2018, 2019). The south-dipping
Woodroffe Thrust has a top-to-the-north sense of shear, and juxtaposes the Fregon
Subdomain in the south (hanging wall) against the Mulga Park Subdomain in the north
(footwall). During the Musgravian Orogeny, the Mulga Park Subdomain attained amphibolite
facies conditions while the Fregon Subdomain reached granulite facies (Camacho and
Fanning, 1995; Scrimgeour et al., 1999; Scrimgeour and Close, 1999), and depleted the rocks
of OH-bearing minerals (Wex et al., 2018; Hawemann et al., 2018).
The Woodroffe Thrust hosts one of the largest occurrences of pseudotachylyte worldwide
(Camacho et al., 1995), but all larger scale shear zones in the hanging wall also show abundant
pseudotachylyte that developed under lower crustal conditions (Camacho, 1997; Hawemann
et al., 2018). Deformation in the Fregon Subdomain associated with the Petermann Orogeny
is concentrated along the sub-eclogitic (~650 °C, 1.2 GPa) Davenport Shear Zone and the
North Davenport Shear Zone (Fig. 1), with little discernible overprint of the earlier granulites
in between (Camacho et al., 1997). The Davenport Shear Zone is a WNW-ESE-striking, strike-
slip zone, with a near horizontal stretching lineation. Deformation inside the Davenport Shear
Zone itself is heterogeneous and strongly localized (Hawemann et al., 2019).

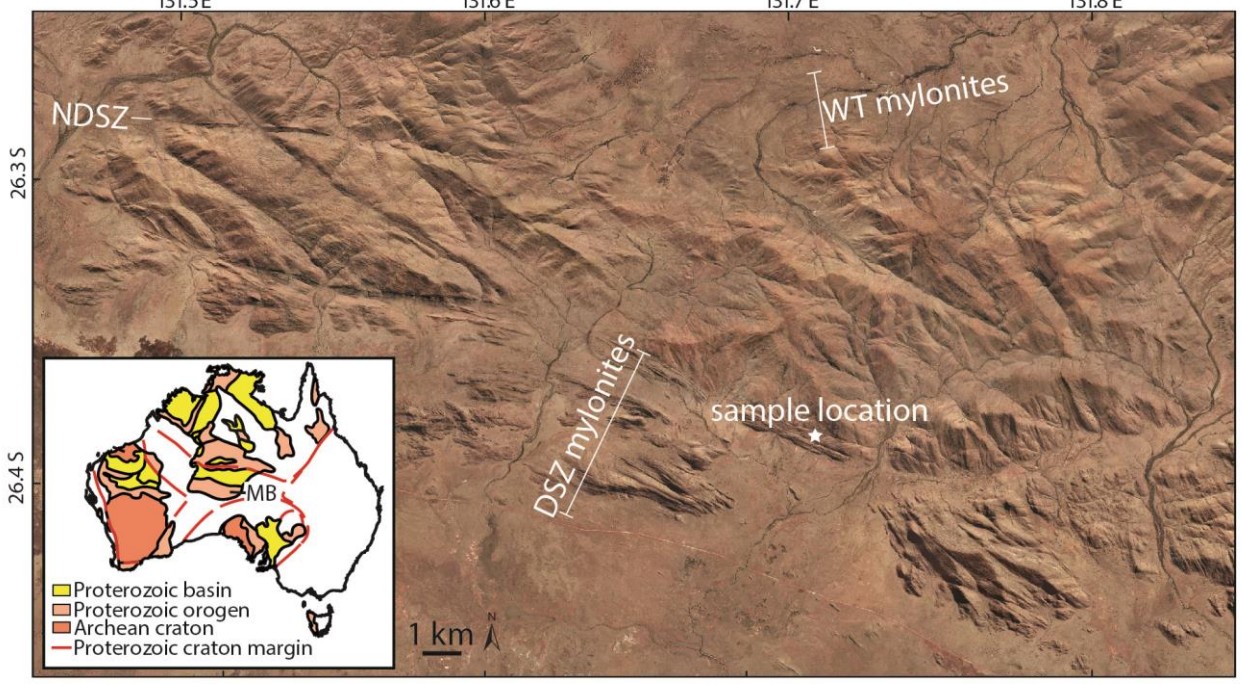


*Figure 1: Airborne imagery of the study area with sample location (26.3849 S, 131.7067 E) in the Davenport Shear Zone (DSZ). NDSZ = North Davenport Shear Zone, WT = Woodroffe Thrust. Image from the Department of Primary Industries and Regions, South Australia (PIRSA), 2012. Inset: Location of the Musgrave Block (MB) in between the amalgamated Australian Cratons. Modified after Evins et al. (2010)*


2.2 Sample description
Fractured garnet is ubiquitous in the Fregon Subdomain and is not exclusively found in
association with pseudotachylyte veins. However, this study focuses on a representative
outcrop for which field relationships, metamorphic, and deformation conditions have been
well established (F68, Hawemann et al., 2018; 26.3849 S, 131.7067 E). This outcrop consists
of a quartzo-feldspathic mylonite with millimetre-sized, granulite facies garnets, and includes
multiple pseudotachylyte veins and breccias. Pseudotachylytes in the studied outcrop are
sheared, as indicated by elongated clasts (Fig. 2a, c), and show the same stretching lineation
as the host mylonite. The original discordant relationship to the host foliation is still
preserved, with the crosscutting relationship most obvious in sections perpendicular to the
stretching lineation (Fig. 2b).
The syn-mylonitic assemblage associated with the Petermann overprint of the felsic
granulites is Qz+Kfs+Pl+Gt+Bt+Ky+Ilm+Rt (mineral abbreviations following Whitney and
Evans, 2010), and is similar to that of the associated sheared pseudotachylyte
(Qz+Kfs+Pl+Gt+Bt+Ky+Rt) (Hawemann et al., 2018). The fine-grained garnet growing within
the pseudotachylyte gives the rock its macroscopic caramel-colour (Fig. 2). Larger fractured
garnets within the granulites are clearly recognizable in polished hand specimens (Fig. 2c) and
are very apparent in thin section (Fig. 3). The metamorphic conditions during shearing of this
pseudotachylye are estimated at ~600 °C and ~1.1 GPa (Fig. 7 of Hawemann et al., 2018).


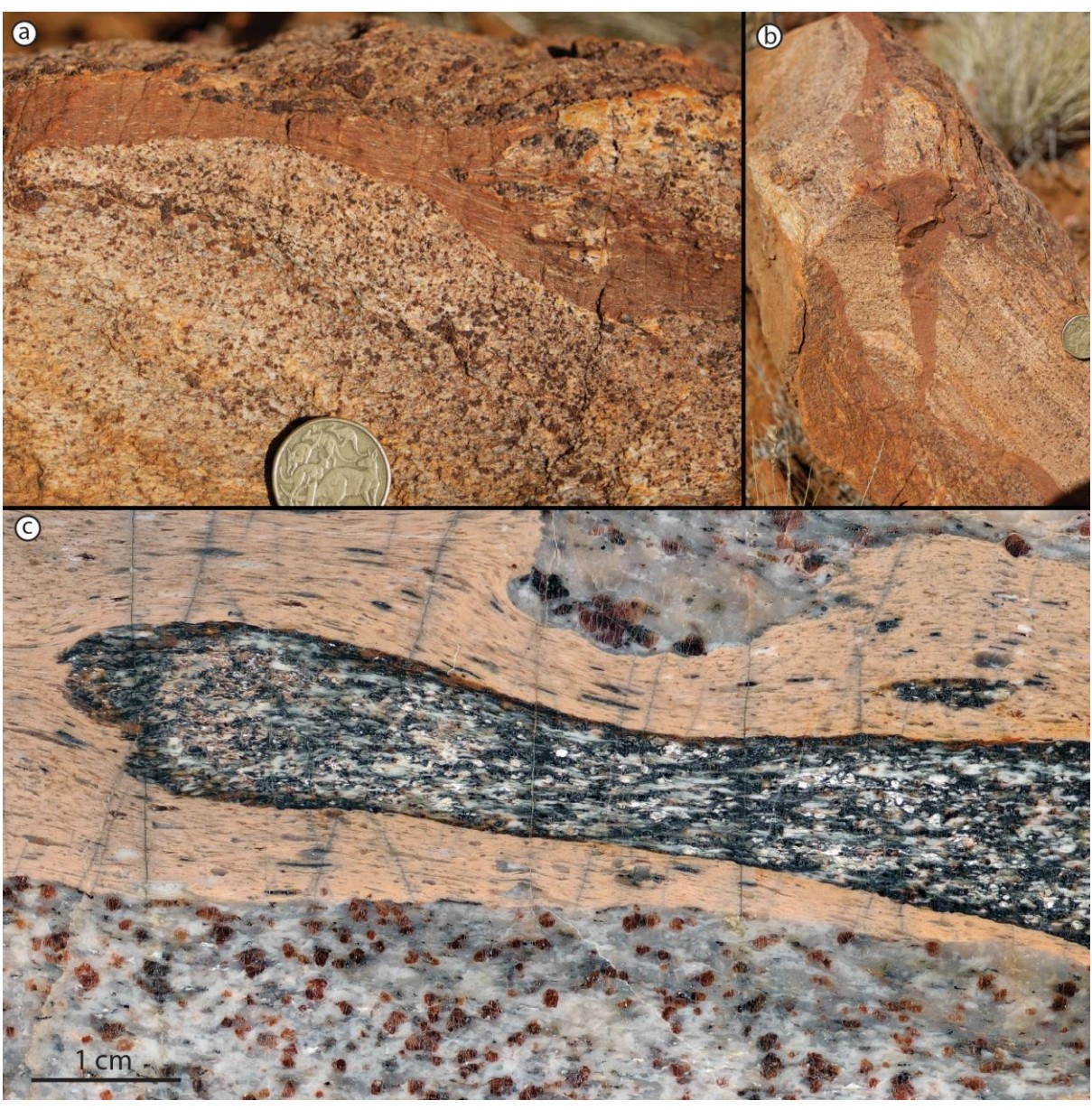

*Figure 2: Sheared pseudotachylyte in a view orthogonal to the foliation of host felsic mylonite, and looking perpendicular (a) and parallel (b) to the stretching lineation. c) Polished hand specimen of a sheared pseudotachylyte breccia with the caramel-coloured foliated pseudotachylyte matrix including elongated clasts and an elongate fragment of mafic granulite. The host rock shows millimetre-sized garnets with fractures. Plane of the polished surface is perpendicular to the foliation and parallel to the stretching lineation.*


## 3 Garnet microstructure and compositional variation

### 3.1 Optical microstructure

Granulite facies garnet porphyroclasts in Musgravian peraluminous gneisses mylonitized during the Petermann Orogeny are almost invariably fractured, irrespective of their proximity to pseudotachylyte (Fig. 3). Large garnet porphyroclasts (>1 mm) are typically slightly elongated with their long axis parallel to the foliation, which is attributed at least partially to resorption. Fractures in garnets often show offsets on the order of a few 100 µm. It is not possible to determine whether these offsets are primarily due to the initial shear fracture or result from subsequent sliding during ongoing ductile shear. Moreover, no consistent sense of shear can be derived from the offsets (Fig. 3a, b). These discrete fractures are sub-planar, commonly have a consistent orientation at a moderate angle to the foliation, and locally occur in conjugate sets (Fig. 3b). Wide fractures are filled with biotite, kyanite and quartz (Fig. 4b). A later generation of unfilled fractures, without any discernible offset, is oriented perpendicular to both the foliation and stretching lineation (Fig. 3b). Garnet porphyroclasts commonly contain rutile exsolution lamellae and inclusions of monazite and kyanite (Fig. A1). The latter are present as aggregates with an overall prismatic shape, possibly representing pseudomorphs after sillimanite (Camacho and Fitzgerald, 2010).

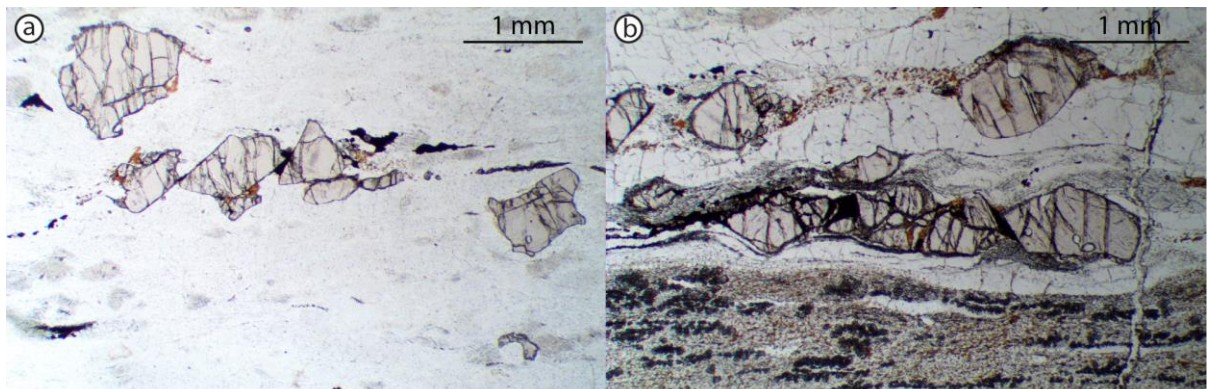

*Figure 3: Thin section photomicrographs in plane polarized light of fractured garnets away from pseudotachylyte (a), and close to sheared and recrystallized pseudotachylyte in the lower part of the figure (b). The dark trails of grains elongated in the foliation of the sheared pseudotachylyte are small new garnets. Section is perpendicular to the foliation and parallel to the stretching lineation.*




164       3.2 Analytical techniques

Quantitative mineral compositions were measured with a JEOL JXA-8200 electron probe
micro-analyzer (EPMA), equipped with a tungsten filament, at the Institute of Geochemistry
and Petrology at ETH Zurich (Switzerland). Natural standards were used for quantification,
and, when available, natural garnet standards were preferred. To reach a spatial resolution
of about 1 μm, an acceleration voltage of 10 kV was set (Fig. 8 in Hofer and Brey, 2007).
Elemental maps were acquired using energy wavelength-dispersive spectrometers in parallel
for calcium, to increase the signal-to-noise ratio. Backscatter electron images (BSE), energy-
dispersive spectrometry (EDS) and electron backscatter diffraction (EBSD) mapping was
carried out on a Quanta 200F field emission gun (FEG) scanning electron microscope at the
ScopeM (Scientific Center for Optical and Electron Microscopy, ETH Zurich). EBSD maps were
collected with an acceleration voltage of 20 kV, a sample tilt of 70° and a working distance of
15 mm. Data were post-processed using chemical indexing with the software OIM 7 by EDAX.
When necessary, three different clean-up techniques were used: neighbour confidence index
correlation, neighbour orientation correlation and grain dilation. Point and map analyses, as
well as BSE images, were combined for correlation with optical microscope images in a QGIS-
project (Open Source Geospatial foundation). Two lamellae were cut with a focused ion beam
(FIB) for transmission electron microscopy (TEM). The microscope used for TEM is a Tecnai
F30 with a FEG source operated at 300 kV and equipped with a Gatan 794 MultiScan CCD
(ScopeM, ETH Zurich).
3.3 Compositional gradients
Granulite facies garnet has a homogeneous composition of $X_{Alm}$ 0.54, $X_{Pyp}$ 0.40, $X_{Grs}$ 0.03, $X_{Sps}$
0.03, whereas garnet neocrystallized during the Petermann Orogeny is more Ca-rich ($X_{Alm}$
0.48, $X_{Pyp}$ 0.28, $X_{Grs}$ 0.22, $X_{Sps}$ 0.02). Grain boundaries of granulite facies garnet and fractures
are decorated with a Ca-enriched rim, 20 to 40 µm wide (Fig. 4c). The length-scale for
variation in Fe ($X_{Alm}$) and Mg ($X_{Pyp}$) is identical to that for Ca ($X_{Grs}$), whereas the Mn content
($X_{Sps}$) does not show any variation (Fig 4d). Neocrystallized garnet is present where the grain
boundary is in contact with, or close to, plagioclase. The outermost rim of remnant garnet has
the same composition as the neocrystallized garnet (Fig. 4d, profile 1). The granulite-facies
plagioclase is partially transformed to a more Na-rich plagioclase with needle shaped
inclusions of kyanite (bottom of Fig. 4e). This reaction provides Ca for the observed diffusion
into garnet (Camacho et al., 2009).
Along fractures across the porphyroclasts, the Ca enrichment is narrower than along the grain
boundaries and the grossular component only reaches up to about $X_{Grs}$ 0.1 (Fig. 4d, profile 2).
Compositional gradients are also present around inclusions in garnet connected to the outer
garnet boundary, providing evidence of Ca diffusion along grain boundaries (right part of Fig.
4c, profile 3 in Fig 4d). Profile 4 (Fig. 4d) was measured next to a kyanite inclusion: the
diffusion length is still comparable to those of profiles 1-3, but Ca concentrations are much
lower. Ca probably diffused along fractures (invisible in the plane of the thin section) towards
the inclusion. In summary, the diffusion length at the original grain boundaries is maximized
where in contact with plagioclase, and otherwise constant at about 20 µm width. However,

variations in diffusion lengths do occur around garnet fragments, without any correlation with the proximity to plagioclase, although the exact relationship in the third dimension is unknown. Surfaces with limited diffusion can often be identified as fracture surfaces, which were exposed to diffusion for a shorter time than original grain boundaries (Fig. 4e). Fractures oriented perpendicular to the foliation and stretching lineation lack any signs of diffusion and are therefore interpreted as later stage extensional fractures.

Some garnets display more complicated compositional patterns, with zones >100 µm of Ca enrichment extending into the porphyroclast's interior, which are not associated with fractures (e.g. the garnet fragment on the far right in Figure 4e). EBSD analysis highlights that the three fragments in the right part of Figure 4e most likely originated from the same grain, as they share a common rotation axis (Fig. 4f). The colours in the inverse pole figure map are not solid, reflecting slight variations of orientation within the crystal. Furthermore, the image quality map shows areas of suppressed Kikuchi patterns (grey value) suggestive of higher dislocation density and therefore possible subgrain boundaries (Fig. 4f). The misorientation angle map (Fig. 4g) reveals a complex pattern of varying crystal orientation (all within the order of 5°) in the fragments, with very distributed zones connected to the edges of the crystal, triangular-shaped zones of misorientation (upper left of Fig. 4g), and discrete zones (lower right of Fig. 4g). The discrete zones of misorientation, about 5 µm wide, correlate well with the Ca-enriched zones (compare Fig. 4e, f, garnet fragment on the right).

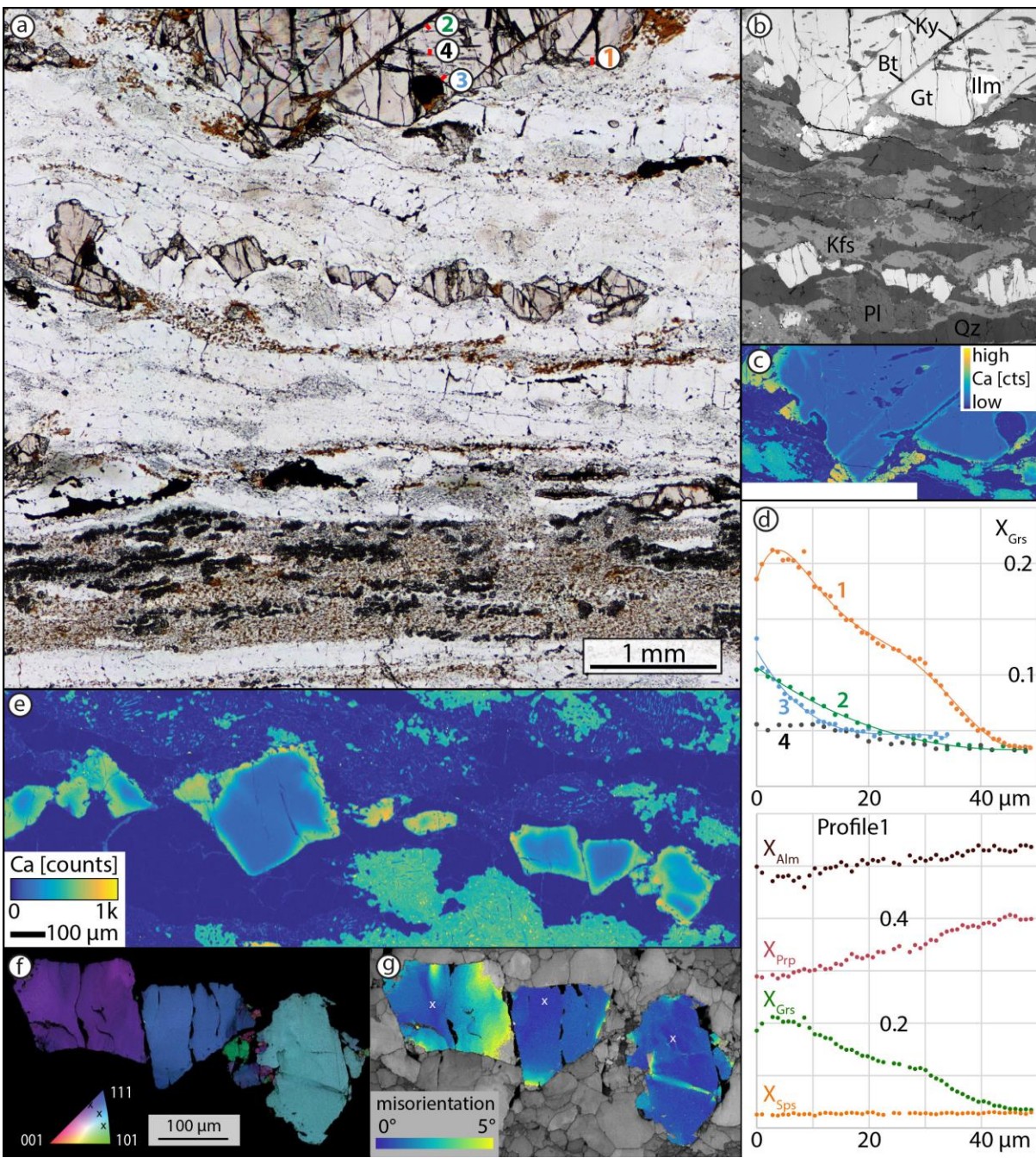


*Figure 4: a) Plane polarized light image of thin section with fractured garnets and a pseudotachylyte vein in the lower part of the image. b) BSE image of the upper area of (a), with same scale as (a). c) EPMA X-ray map for Ca reveals an enrichment in thin gradational rims along grain boundaries and fractures, and within neocrystallized garnet (euhedral, orange). d) Grossular component profiles indicated on (a) (profile lines are not to scale for the sake of visibility) and compositional profiles for four garnet end-members in profile 1. e) EPMA X-ray map for Ca for the garnet fragments in the center of (a). Note the uneven colours in the plagioclase and the blue kyanite needles. f) Inverse pole figure map with superimposed image quality map for garnet fragments shows a common rotation pole. g) Misorientation map relative to reference point for each fragment reveals internal lattice distortions.*


3.4 Texture of deformed garnets
Two to three orientations of fractures are generally present in a single garnet crystal and
coincide with the trace of the (101)-plane derived from EBSD data (Fig. 5a, b). Fracture set (I)
in the example of Figure 5a is often associated with a relative rotation of both sides, as visible
from the difference in colour. In the lower part of the grain, where the fracture density is very
high, more subgrains are present. The subgrain spatial density increases towards the original
grain boundary and some subgrains are "eroded" by ductile shearing and strung out along the
foliation. This demonstrates that ductile shearing outlasted subgrain formation and
fracturing. The fractures described above are all crosscut by extensional fractures (set II in Fig.
5a), oriented perpendicular to the stretching lineation and foliation, which do not show any
associated distortion of the crystal lattice.
The garnet porphyroclast of Figure 5c shows a central fracture as well as a set of two other
parallel fractures. The central fracture is the only one with significant offset and is filled with
kyanite and quartz. This fracture displays misorientations of more than 5° towards the right-
hand side of the scan, but none towards the left-hand side. In the lower left corner of the
fragment, subgrains are observed with misorientations, relative to the average orientation,
typically in the range of 10°. Misorientation axes are often parallel to (111) and (101). The
lowermost fragment shows a wide zone of progressive rotation. The chemical profile in Figure
5e shows the highest Ca counts towards the boundaries of the porphyroclasts and, internally,
towards two fractures. The larger fracture with apparent offset of the two garnet fragments
exhibits a less well-developed zone of Ca enrichment when compared to the tight fracture
with introduced lattice distortion.

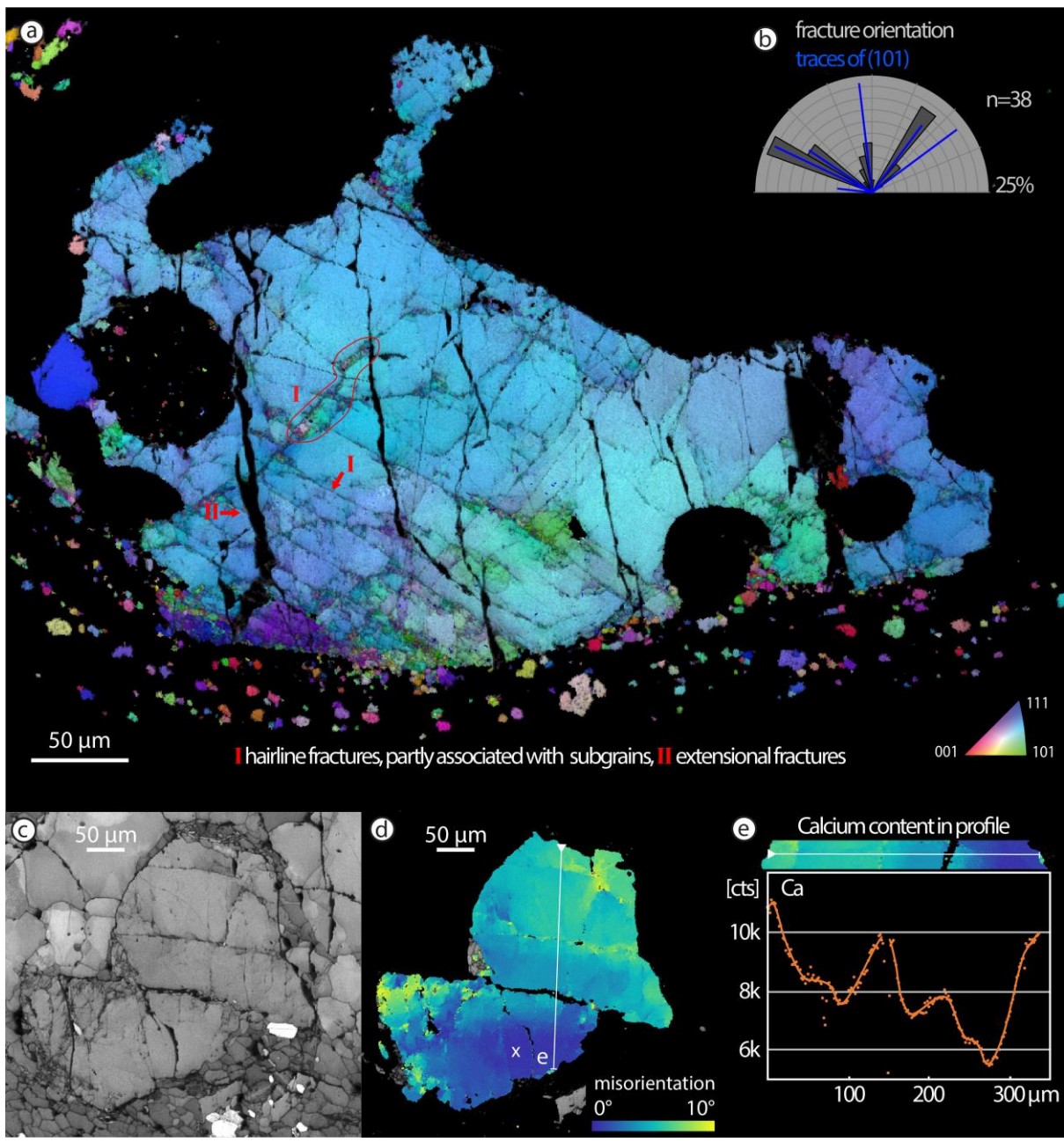


*Figure 5: a) Inverse pole figure map of fractured garnet with three dominant orientations of fractures. b) Rose diagram correlating traced fracture orientations and (101)-planes for garnet in (a). c) Image quality map of a fragmented garnet with subgrains. d) Misorientation plot (with respect to the point marked with the white x) shows long wavelength bending in the lower fragment and distortion in the crystal lattice induced by a fracture in the upper fragment. e) EDS-calcium counts for the profile marked as a thin white line in (d).*



3.5 TEM investigations
The garnet fragment of Figure 4g was further investigated using TEM, as it includes a narrow
zone of misorientation without fractures and is therefore suitable for preparation of FIB-
lamellae. As visible in Figure 6a (around profile 1), the image quality map shows a well-defined
narrow, darker grey band, possibly indicating high dislocation density. The zone is even more
evident in the misorientation plot (Fig. 6b) and changes from about 5 µm wide, with discrete
boundaries to the right, to a wider (> 10 µm) band towards the left of the image. In the upper
left part of the image, a subgrain boundary with > 5° misorientation transitions into a zone of
gradual misorientation. The misorientation axis is consistently parallel to (101) with minor
rotation around (111) (Fig. 6c, Fig. A2). Misorientation profiles reveal a slight asymmetry
within the narrow band, where the lower boundary appears to be sharper. Misorientation
changes more gradually within the wider portion of the misorientation band. Locally,
subgrains developed with discrete boundaries, documenting a misorientation of usually
around 5-10° (profile 3 in Fig. 6d). The FIB-lamella was cut across the narrow band of
misorientations (Fig. 6e). The lower boundary corresponds to a narrow discrete zone, without
visible dislocations (Fig. 6f). The upper boundary is marked by a series of dislocation walls and
only a few free dislocations are visible, which are often organized in arrays (Fig. 6g, h). The
existence of dislocation walls and subgrain boundaries indicates recovery by dislocation climb
(e.g., Hobbs, 1968; Passchier and Trouw, 2005).

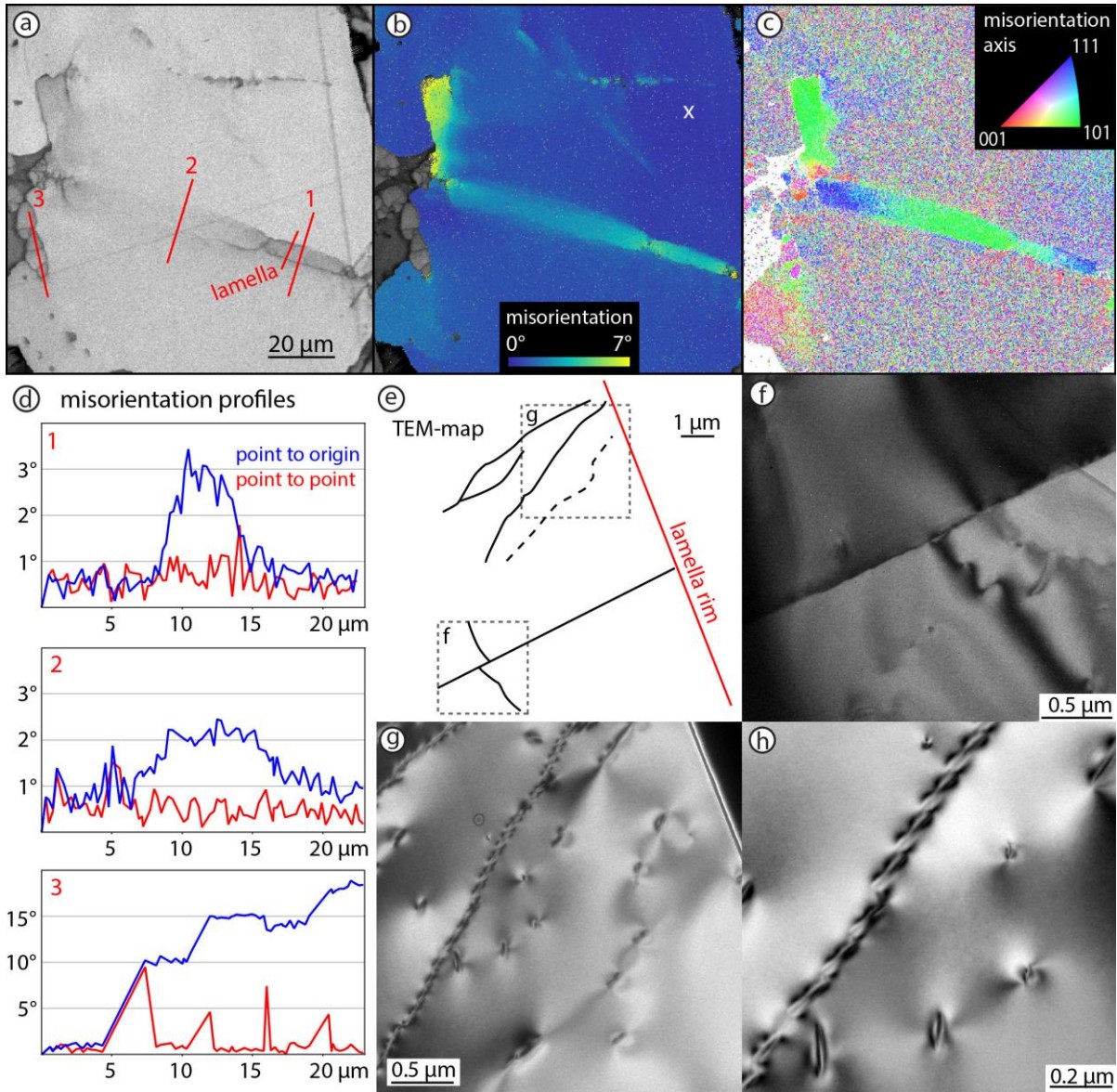

*Figure 6: a) Image quality map of the garnet fragment (compare Fig. 4f) with darker zones that can be interpreted as areas of high dislocation density and location of the FIB-lamella. b) Misorientation plot with respect to the reference point (marked with the white x) shows a discrete zone of misorientation, which has discrete boundaries in the right part of the image, but is more distributed towards the left. c) Misorientation axis plot with respect to the average orientation of the grain shows a consistent rotation around the (101) and (111) axes. For pole figure plots, see Fig. A2. d) Misorientation profiles indicated in a), for (1) the narrow zone, (2) the more distributed zone and (3) for subgrains. e) Overview sketch of the FIB-lamella used for TEM-analysis for correlation with the EBSD data. f) Sharp contrast boundary in the lower part of the lamella. g) Two dislocation walls with a few free dislocations, which are partly linking up parallel to the dislocation walls. h) Detail of the centre of (g)*

**4  Discussion**

Garnets in this study show evidence for both brittle and ductile deformation under relatively low temperatures of about 600 °C, as inferred from synchronous diffusion and ductile shearing of pseudotachylyte (Hawemann et al, 2018). This is below the experimentally determined values for the onset of crystal-plastic deformation of garnet (Wang and Ji, 1999) at the higher strain rates considered typical of mylonitic shear zones (> $10^{-14}$ s$^{-1}$). In contrast to experiments, many natural examples (Vollbrecht et al., 2006; Bestmann et al., 2008; Austrheim et al., 2017) indicate crystal plasticity of garnet at lower temperatures between 650 °C and 700 °C.

The presence of microstructures and textures consistent with dislocation climb and recovery, as well as subgrain rotation, in garnet at around 600 °C is in agreement with previous studies (Bestmann et al., 2008; Massey et al.,2011). No evidence for grain boundary sliding is observed, since subgrains show rotation around a specific crystallographic axis. Rotation around (111) and (101) is in accordance with the slip systems described by Voegelé et al. (1998).

Multiple generations of overprinting fractures with different orientation demonstrate repeated fracturing events. Extensional fractures do not show any induced lattice distortion or diffusion and therefore occurred after the temperature had decreased to values too low for diffusion (Camacho et al., 2009), possibly during exhumation (compare Prior, 1993 and Ji et al., 1997).

In contrast to the observations of Austrheim et al. (2017), Papa et al. (2018) and Petley-Ragan et al. (2019) from other examples in the deep continental crust, no "explosive fracturing", "shattering" or "fragmentation" of garnet is observed in relict porphyroclasts immediately

adjacent to pseudotachylyte. The fractures described here are generally planar and often
consistently oriented, in some cases showing single and conjugate shear offsets. Fractured
garnet is not restricted to the boundary with pseudotachylyte and is still present even in
samples without pseudotachylyte, where the nearest pseudotachylyte is possibly many
metres or more away. Fracturing in this case cannot be related to thermal shock (Papa et al.,
2018) or localized high stress due to (seismic) fracture propagation (Austrheim et al., 2017;
Petley-Ragan et al., 2019), but must reflect a larger scale distribution of differential stresses
in the lower crust that were, at least transiently, high enough to cause brittle garnet failure
(Hawemann et al., 2019). This could be due to stress pulses from earthquakes in the shallower
brittle regime (Trepmann and Stöckhert, 2002; Ellis and Stöckhert, 2004; Jamtveit et al.,
2018a, b; Jamtveit et al., in press) or a more local, lower crustal source due to jostling of less-
deformed strong blocks within an irregular shear zone network (Hawemann et al., 2019).
The narrower Ca diffusion profiles on some fractures relative to garnet rims and crosscutting
relationships suggest that fracturing was recurrent under sub-eclogite facies metamorphic
conditions, as also indicated by the occasional presence of kyanite in some fractures. The
presence of kyanite needles and the absence of zoisite/clinozoisite or epidote, as a
breakdown product of plagioclase during sub-eclogitic metamorphism (Fig. 3b), indicate
relatively dry lower crustal conditions (Hawemann et al., 2018). According to Wayte et al.
(1989), this indicates a water activity of < 0.004, calculated for rocks of comparable
composition and P-T conditions. However, new biotite did form in fractures across relict
garnet, so conditions were probably not strictly anhydrous. The sheared and recrystallized
pseudotachylyte developed a similar synkinematic assemblage as the host mylonite,
demonstrating that there is also no marked partitioning of water into the frictional melt,
which implies little free or bound water available in the original source rock (e.g. Wex et al.,
2018). The effect of pore-fluid pressure on the effective confining pressure must therefore
have been negligible.
As reported in Hawemann et al. (2019), the dynamically recrystallized quartz grain size and
microstructure in the host rock mylonites indicates that long-term flow stresses were not
particularly high, on the order of less than 10 MPa. The ambient pressure of ca. 1.1-1.2 GPa
determined for the host rocks should therefore be close to the lithostatic value (Mancktelow,
2008). Figure 7 shows a simple linear plot of the Mohr-Coulomb failure criterion for an angle
of internal friction of 30° (coefficient $\mu$ = 0.6), a lithostatic load of 1.2 GPa, and no pore fluid
pressure. This plot is only qualitative, since the angle of internal friction could decrease
towards higher pressure (Shimada et al., 1983 ). However, the summary of experimental
results in Byerlee (1978) indicates that there may be little change at least up to pressures
similar to those considered here. It follows that the differential stress for fracture initiation
must have been of the same order as the confining pressure (Fig. 7). As discussed in detail in
Hawemann et al. (2019), such high differential stresses, leading to garnet fracture and the
development of abundant pseudotachylyte, can only have been transient and presumably
related to repeated short-term seismic events in the lower continental crust (Hawemann et
al., 2018; Jamtveit et al, 2018a, b; Menegon et al., 2017). The lack of shattered garnet adjacent
to pseudotachylyte in these samples may reflect drier conditions relative to those in the
Bergen Arc (Austrheim et al., 2017) and Mont Mary (Papa et al., 2018). The samples studied
could therefore represent one end-member of the lower continental crust, where
deformation occurs without the initial presence or influx of free water during fracturing and
subsequent crystal-plastic deformation.

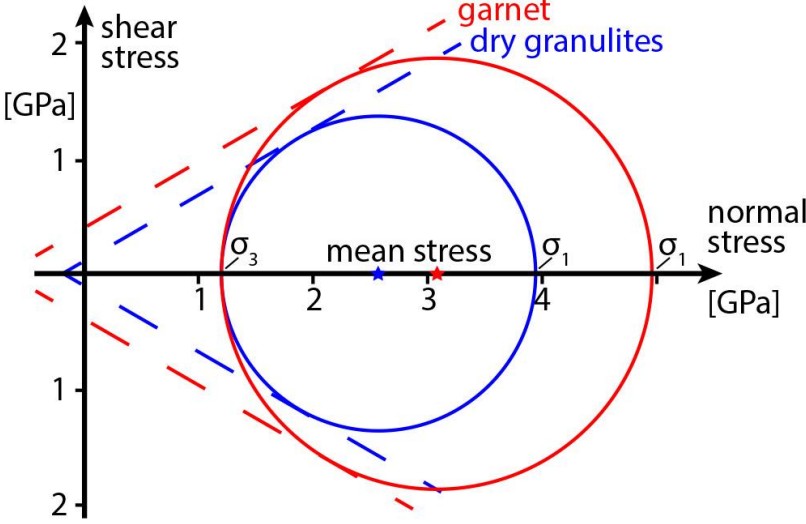

*Figure 7: Mohr circles for fracturing of dry granulites and garnet at 1.2 GPa lithostatic load*

## 5 Conclusions

In dry lower continental crust deformed under conditions of ca. 600 °C and 1.1 GPa, garnet shows both single and conjugate sets of shear fractures, fractures with associated subgrains and induced lattice damage around fractures, subgrain formation without fracturing, and late-stage extensional fractures. Most of these fractures show a strong crystallographic control, with fracturing preferentially occurring along the (101) planes of garnet. Dynamic recrystallization is evident from inferred subgrain rotation recrystallization and recovery is manifested by the presence of dislocation walls. The observed microstructures of garnets are interpreted to record transient high stresses during deep seismic events in the lower crustal Fregon Subdomain. This is also indicated by the abundant occurrence of pseudotachylyte developed under similar lower crustal conditions and, possibly, by the variability of recrystallized quartz grain sizes including values down to a few micrometres (Hawemann et al. 2009b). The studied example represents one end-member of lower continental crustal behaviour where, because of earlier metamorphic dehydration and the intracratonic position

well removed from the plate margin, rocks were initially dry and water was not introduced
during fracturing and crystal-plastic deformation.

**Author contributions**

All authors listed took part in at least two of the three field seasons. NM assisted FH in the data
collection and interpretation. AC's and GP's knowledge in the field of garnet deformation and diffusion
processes were crucial in preparing the manuscript. SW contributed to the microprobe and SEM work.
FH prepared the manuscript with contributions from all co-authors.

**Competing interests**

The authors declare that they have no conflict of interest.

**Acknowledgements**

We want to thank Matthias Konrad-Schmolke and an anonymous reviewer for their critical
comments which improved the manuscript. We gratefully acknowledge permission granted to
work on the Anangu Pitjantjatjara Yankunytjatjara Lands (APY) to carry out our field work in
the area. The Northern Territory Geological Survey (NTGS) and Basil Tikoff (Department of
Geoscience, University of Wisconsin) are thanked for their logistical support and the Nicolle
family of Mulga Park station for their hospitality. The Scientific Center for Optical and Electron
Microscopy (ScopeM) provided the facilities for the scanning electron microscopy work, and
help by Karsten Kunze, Luiz Morales and Fabian Gramm is especially acknowledged. Luca
Menegon is thanked for his review of the first author's doctoral thesis. This project was
financed by the Swiss National Science Foundation (SNF) grant 200021_146745 and by the
University of Padova (BIRD175145/17: The geological record of deep earthquakes: the
association pseudotachylyte-mylonite).

## Data Availability

All data used in this paper can be accessed through the depository of the Open Science
Framework here: https://osf.io/yrzgh/

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

**Appendix**

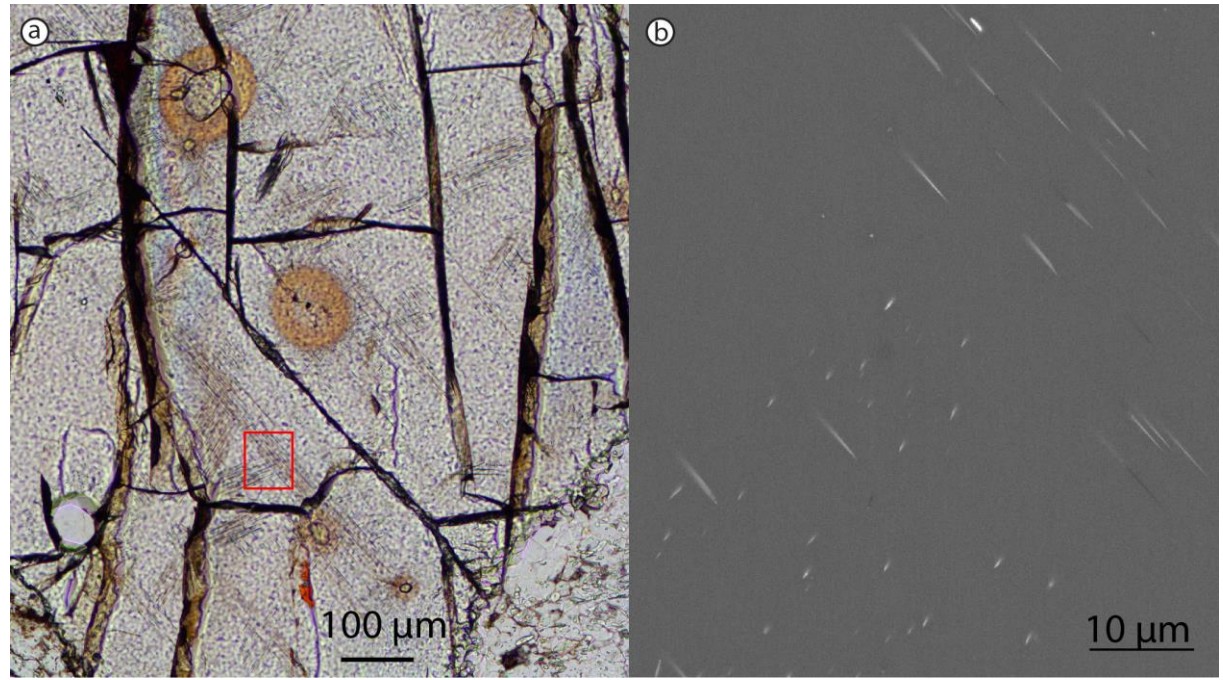


*Figure A1: Thin section image in plane polarized light of a garnet crystal with monazite inclusions (with halos) and rutile-exsolution needles. b) BSE-image of the area indicated with the red box.*




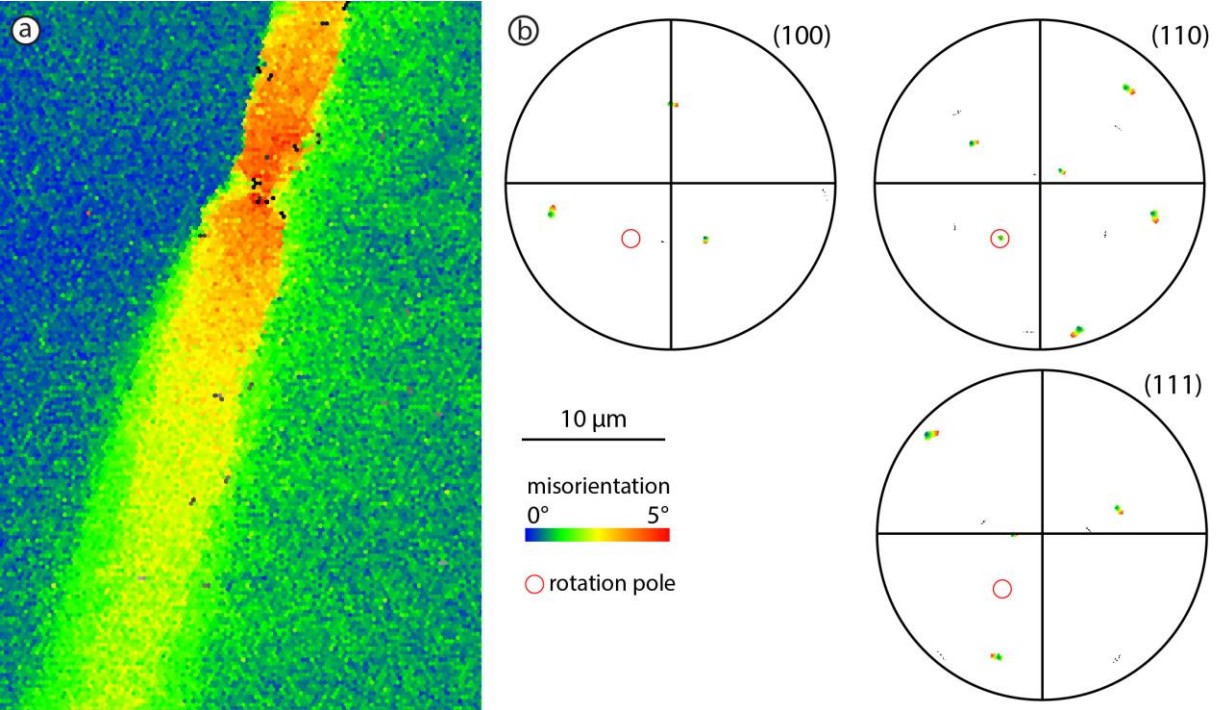


*Figure A2: a) Misorientation map-detail for Fig. 6b), with b) pole figure plots for garnet axis*

*with the same colour scheme. The plots reveal a rotation around a (101)-axis, as indicated by*

*the red circle.*