# Peer review of "Fracturing and crystal plastic behaviour of garnet under seismic stress in the 1 dry lower continental crust (Musgrave Ranges, Central Australia) 2 3 4 5 Friedrich Hawemann1\*, Neil Mancktelow1, Sebastian Wex1, Giorgio Pennacchioni2, Alfredo Camacho3<"

_Solid Earth, 2019_

## Referee Comment (RC1) · Matthias Konrad-Schmolke (Referee) · 15 Jul 2019

Review Fracturing and crystal plastic behavior of garnet under seismic stress in the dry lower continental crust (Musgrave Ranges, Central Australia) Authors Friedrich Hawemann, Neil Mancktelow, Sebastian Wex, Giorgio Pennacchioni, Alfredo Camacho Summary The authors of the manuscript present detailed data on plastic garnet deformation from a medium pressure/medium temperature shear zone from the Musgrave Block in Southern Australia. The investigated felsic samples stem from the Davenport Shear Zone with a strike slip sense of shear and experienced granulite facies metamorphism prior to deformation, which resulted in an intense dehydration of the rocks.

[Figure]

The authors interpret this intense dehydration and the lack of infiltrating fluids during the deformation to be the reason for the rheologically distinct properties of the rocks. These distinct rheological properties involve the crystal-plastic deformation of garnet at temperatures in the order of 600°C. Interestingly, such temperatures are significantly lower than those experimentally determined for crystal-plastic deformation in garnet. The authors therefore draw several conclusions from their observations: (1) the experimental data for crystal plasticity in garnet cannot be transposed to natural conditions, (2) the differential stresses causing the plastic deformation of garnet are high, transient and caused by lower crustal earthquakes.

General comment The manuscript focuses on a geoscientifically relevant and interesting topic, i.e. paleo-earthquake features in deformed rocks and their interpretation with respect to deformation mechanisms and stress quantification. The authors write in concise English and the structure of the manuscript allows the reader to follow the authors' arguments and discussion easily. The topic of the manuscript is up to date and the data presented here is all new. However, I think that some of the interpretations in the authors' discussion are not fully supported by the data. One crucial argument for crystal plasticity in garnet is the observation of dislocation walls that mark the boundary of one subgrain in the garnet crystals. The authors state that these dislocation walls are the result of dislocation climb in the crystal (lines 252-253) and therefore indicate the activity of viscous deformation mechanisms in garnet. I am not entirely convinced that these dislocation walls are only produced by the migration of dislocations through the crystal, although I am not aware of studies that demonstrate neither pro nor contra arguments. The fact that the authors do not cite any references is also not helpful with this regard. However, there is evidence that such dislocation walls can be generated in undeformed rocks, e.g. during fluid infiltration, such as demonstrated in Konrad-Schmolke et al., 2018. Of course, fluid infiltration does not play a role in the rocks presented here, but other mechanisms for the formation of the dislocation walls must be discussed in this manuscript, as these structures are a fundamental argument for crystal plasticity. Furthermore, the interpretation of the presence of rotated

subgrains in terms of subgrain rotation recrystallization is, in my opinion, also questionable. Konrad-Schmolke et al., 2007 demonstrate the presence of subgrains in garnets (and their slight misorientations) in undeformed rocks. In general, I think that the manuscript would very much benefit from a more indepth discussion of these features. The papers cited in this review should only serve as examples and I think that there are many other contributions to the topic that I am not aware of at the moment. However, I think the manuscript is very well suitable for publication after moderate revisions and a more thorough discussion.

Specific comments

Line 138: if the fractures a dilatant there must be some material in the cracks. Can that be evaluated?

What about the other, fast diffusing elements, such as Mn and Mg? Differences in diffusion lengths would indicate different diffusion velocities and thus support the idea of a diffusional modification.

Lines 196-197: This diffusional modification is likely due to subgrain boundaries that might or might not be associated with subgrain rotations. This has been demonstrated in Konrad-Schmolke et al., 2007 (EJM). This should be discussed or at least mentioned.

References cited in this review

Konrad-Schmolke, M., O'Brien, P.J. & Heidelbach, F. (2007). Compositional reequilibration of garnet: the importance of sub-grain boundaries. European Journal of Mineralogy, 19, 4, 431-438. Konrad-Schmolke, M., Halama, R., Wirth, R., Thomen, A., Klitscher, N., Morales, L., Schreiber, A. & Wilke, F. D. (2018). Mineral dissolution and reprecipitation mediated by an amorphous phase. Nature Communications, 9(1), 1637.

---

## Referee Comment (RC2) · Anonymous Referee #2 · 19 Jul 2019

Summary: The study by Hawemann et al. presents naturally deformed garnet grains that show micro- and nanostructures interpreted as brittle and crystal plastic deformation features. The main point the authors make is that the observed crystal plastic deformation occurred at 'dry' conditions and at around 600 °C contrasting laboratory data showing the onset of plastic flow of garnet at T > 850 °C. The manuscript is very well written and the subject presented is interesting. Therefore, I suggest publication in Solid Earth after moderate revisions.

General comments: The main message of this manuscript is the occurrence of crystal plasticity in garnet at temperatures well below the laboratory derived data for the

[Figure]

onset of crystal plastic deformation in garnet. The authors, therefore, state that laboratory data fail to explain the natural observations. Of course laboratory experiments are most often very simplified rendering extrapolation to natural systems rather challenging. Though, I miss a bit the explanation why the laboratory data does not match natural observations. Is it because the samples in the laboratory were even drier than the natural rock delaying the onset of crystal plastic deformation in the laboratory? Obviously there were some fluids present due to the occurrence of biotite. Though in some parts of the manuscript, the authors state that a Ca-rich garnet forms instead of epidote, because of the low water activity (line 290). Perhaps there was enough water around to facilitate crystal plastic deformation but not enough to stabilize epidote? I think it would improve the manuscript to discuss the role of fluids on crystal plastic deformation in more detail. This might also explain the discrepancy between laboratory data and the natural observations.

Detailed comments: Line 71: In this context I think crystal plastic deformation instead of ductile deformation is more appropriate. Lines 276-278: Did you investigate/find garnet crystals that were cut by a pseudotachylyte? Both studies that you cite, Austrheim et al. (2017) and Papa et al. (2018), demonstrate garnet crystals that are situated right next to a pseudotachylyte-bearing fault. I mention this, because as strain rate and stresses decrease very rapidly with increasing distance, the required stresses and/or strain rates at a few mm to the fault might not be sufficient anymore to extensively fragment garnet. Line 286: Delete 'of some'. Lines 291-292: So everything is dry, but suddenly there is biotite? You should discuss the presence/absence of hydrous minerals a bit more. Lines 304-305: Shimada et al. (1983) experimentally investigated that the angle changes from around 30 to approx. 45° with increasing pressure. Lines 311-313: See comment above. As water seems important you should perhaps quantify the amount of water? There is some water present in the other field studies mentioned, but not very much. How should the presence of a fluid help to fragment the rock? Figure 5: The difference between fracture types I and II is not very clear to me. The magnification at which the image was taken is quite low and therefore it is difficult to

see subgrains.

References cited in this review:

Austrheim, H., Dunkel, K.G., Plümper, O., Ildefonse, B., Liu, Y., Jamtveit, B., 2017. Fragmentation of wall rock garnets during deep crustal earthquakes. Sci. Adv. 3, 1–7. doi:10.1126/sciadv.1602067

Papa, S., Pennacchioni, G., Faccenda, M., 2018. Thermal fragmentation of garnet during deep-seated co-seismic frictional heating Thermal fragmentation of garnet during deep-seated co-seismic frictional 46, 471–474.

Shimada, M., Cho, A., Yukutake, H., 1983. Fracture strength of dry silicate rocks at high confining pressures and activity of acoustic emission. Tectonophysics 96, 159–172. doi:10.1016/0040-1951(83)90248-2

---

## Author Comment (AC1) · 20 Aug 2019

We want to thank Matthias Konrad-Schmolke for his critical review and suggestions. Below is a list of all comments from the reviewer (RC), answers from the authors (AC) and manuscript changes (MC).

—

Reviewer 1, general comment RC: One crucial argument for crystal plasticity in garnet is the observation of dislocation walls that mark the boundary of one subgrain in the garnet crystals. The authors state that these dislocation walls are the result of dislo-

cation climb in the crystal (lines 252-253) and therefore indicate the activity of viscous deformation mechanisms in garnet. I am not entirely convinced that these dislocation walls are only produced by the migration of dislocations through the crystal, although I am not aware of studies that demonstrate neither pro nor contra arguments. The fact that the authors do not cite any references is also not helpful with this regard. However, there is evidence that such dislocation walls can be generated in undeformed rocks, e.g. during fluid infiltration, such as demonstrated in Konrad-Schmolke et al., 2018. Of course, fluid infiltration does not play a role in the rocks presented here, but other mechanisms for the formation of the dislocation walls must be discussed in this manuscript, as these structures are a fundamental argument for crystal plasticity. Furthermore, the interpretation of the presence of rotated subgrains in terms of subgrain rotation recrystallization is, in my opinion, also questionable. Konrad-Schmolke et al., 2007 demonstrate the presence of subgrains in garnets (and their slight misorientations) in undeformed rocks. In general, I think that the manuscript would very much benefit from a more indepth discussion of these features. The papers cited in this review should only serve as examples and I think that there are many other contributions to the topic that I am not aware of at the moment. However, I think the manuscript is very well suitable for publication after moderate revisions and a more thorough discussion.

AC: As noted by the reviewer, fluid infiltration does not play a role in the rocks presented here, so the mechanism proposed in Konrad-Schmolke et al., 2018 cannot be relevant in this case. The rocks we are considering here are also clearly deformed, with stresses being high enough (at least transiently) to cause fracturing of garnet. Progressive subgrain rotation by migration of dislocations into walls bounding such subgrains is a mechanism that has been very widely proposed both in the material and earth sciences. There is a large body of published work supporting and describing this mechanism – indeed as noted by the reviewer "I think that there are many other contributions to the topic". It is not the aim of the current manuscript to provide an exhaustive review be we have now added the following additional references as

a representative selection: Hobbs, B.E.: Recrystallisation of single crystals of quartz. Tectonophysics, 6, 353-401, 1968. Passchier, C.W., Trouw, R.A.J.: Microtectonics (2nd Edition), Springer, Heidelberg, 366 pp., 2005.

—

Reviewer 1, specific comments

RC: Line 138: if the fractures a dilatant there must be some material in the cracks. Can that be evaluated?

AC: No, these fractures remain empty, as implied by the word "unfilled" in the original text. These fractures are Mode 1 extensional fractures, which we think open during propagation of the seismic wave and immediately close, preventing any mineral filling.

MC: The word "dilatant" is perhaps better replaced with "extensional", so the text now reads "An apparent late generation of unfilled extensional fractures [. . .]". All other similar references to "dilatant fractures" have also now been changed to "extensional fractures".

-

RC: What about the other, fast diffusing elements, such as Mn and Mg? Differences in diffusion lengths would indicate different diffusion velocities and thus support the idea of a diffusional modification.

AC: In Figure 4 d) we present the profiles for Fe, Mg, Mn. Fe and Mg show the same diffusion length as Ca. Mn does not show any measurable modification throughout the crystal.

MC: This observation was missing in the text, therefore we added the following sentence for clarification: "The length-scale for variation in Fe (XAlm) and Mg (XPyp) is identical to that for Ca (XGrs), whereas the Mn content (XSps) does not show any variation (Fig 4d)."
-

RC: Lines 196-197: This diffusional modification is likely due to subgrain boundaries that might or might not be associated with subgrain rotations. This has been demonstrated in Konrad-Schmolke et al., 2007 (EJM). This should be discussed or at least mentioned.

AC: Since a subgrain is defined by a relative crystallographic rotation (commonly taken arbitrarily as between ca. 4° and 15°, when it is considered to be a "high-angle boundary" to a "new grain", e.g. Urai et al., 1986), the generally accepted argument is that subgrain boundaries are always associated with subgrain rotations, as new dislocations are continuously added to the subgrain boundaries (e.g. Passchier and Trouw, 2005, p.43). We have added the reference to Konrad-Schmolke et al. (2007), as well as recent publications of Petley-Ragan et al. (2019), Jamtveit et al. (2018a,b, in press), Engi et al. (2018), Giuntoli et al. (2018) and Angiboust et al. (2017) when comparing and contrasting our "dry" results to fracture and diffusion in garnets in deep-seated rocks where fluid infiltration plays an important role.

---

## Author Comment (AC2) · 20 Aug 2019

We want to thank the anonymous reviewer for his critical review and suggestions. Below is a list of all comments from the reviewer (RC), answers from the authors (AC) and manuscript changes (MC).

—

Reviewer 2, general comment

RC: The main message of this manuscript is the occurrence of crystal plasticity in garnet at temperatures well below the laboratory derived data for the onset of crystal

[Figure]

plastic deformation in garnet. The authors, therefore, state that laboratory data fail to explain the natural observations. Of course laboratory experiments are most often very simplified rendering extrapolation to natural systems rather challenging. Though, I miss a bit the explanation why the laboratory data does not match natural observations. Is it because the samples in the laboratory were even drier than the natural rock delaying the onset of crystal plastic deformation in the laboratory? Obviously there were some fluids present due to the occurrence of biotite. Though in some parts of the manuscript, the authors state that a Ca-rich garnet forms instead of epidote, because of the low water activity (line 290). Perhaps there was enough water around to facilitate crystal plastic deformation but not enough to stabilize epidote? I think it would improve the manuscript to discuss the role of fluids on crystal plastic deformation in more detail. This might also explain the discrepancy between laboratory data and the natural observations.

AC: We have added text in several places to expand the discussion of the apparent discrepancy with laboratory data, in particular considering the potential effects of strain rate and role of fluids. We agree that this should have been treated in more detail, which is why we now have a rather more nuanced approach, considering factors that may have an influence rather than just stating that there is a difference.

—

Reviewer 2, specific comments

RC: Line 71: In this context I think crystal plastic deformation instead of ductile deformation is more appropriate.

AC: agree

MC: changed sentence: "between brittle and crystal plastic deformation of garnet"

-

RC: Lines 276-278: Did you investigate/find garnet crystals that were cut by a pseudotachylyte? Both studies that you cite, Austrheim et al. (2017) and Papa et al. (2018), demonstrate garnet crystals that are situated right next to a pseudotachylyte-bearing fault. I mention this, because as strain rate and stresses decrease very rapidly with increasing distance, the required stresses and/or strain rates at a few mm to the fault might not be sufficient anymore to extensively fragment garnet.

AC: In the text, we clearly state that "Granulite facies garnet porphyroclasts in Musgravian peraluminous gneisses mylonitized during the Petermann Orogeny are almost invariably fractured, irrespective of their proximity to pseudotachylyte (Fig. 3)." This is different than what was observed in the examples of Austrheim et al. (2017) and Papa et al. (2018) mentioned above, which is why we made such a clear statement originally. On the basis of this observation, we argue in the text that the whole rock was affected by high stresses during transient seismic events and that garnet fracturing is not restricted to the localized damage zone of a propagating fracture (Petley-Ragan et al, 2019; Austrheim et al., 2017) or thermal shock immediately adjacent to the high temperature pseudotachylyte (Papa et al., 2018).

-

RC: Line 286: Delete 'of some'.

AC: agree

MC: Typing error corrected.

-

RC: Lines 291-292: So everything is dry, but suddenly there is biotite? You should discuss the presence/absence of hydrous minerals a bit more.

AC: Biotite is a typical mineral of granulite facies assemblages up to the point of melting (with biotite then providing the water for "anhydrous" melting) and even then biotite is a common mineral in the restite assemblage. "Kinzigite", which is a typical "dry" lower-crustal granulite facies rock, is actually defined as having garnet + biotite. As

noted by Pennacchioni and Cesare (1997), under upper amphibolite facies conditions, newly grown biotite can actually act as a sink for any free water available and the same will be true for the "dry" high pressure upper amphibolite ("sub-eclogitic") facies conditions relevant to the current study. Pennacchioni, G. & Cesare, B., 1997. Ductile-brittle transition in pre-Alpine amphibolite facies mylonites during evolution from water-present to water-deficient conditions (Mont Mary nappe, Italian Western Alps). Jour. Metm. Geol. 15, 777-791.

-

RC: Lines 304-305: Shimada et al. (1983) experimentally investigated that the angle changes from around 30 to approx. 45âŮę with increasing pressure.

MC: The reference was added to the text: "This plot is only qualitative, since the angle of internal friction could decrease towards higher pressure (Shimada et al., 1983)."

-

RC: Lines 311-313: See comment above. As water seems important you should perhaps quantify the amount of water? There is some water present in the other field studies mentioned, but not very much. How should the presence of a fluid help to fragment the rock?

AC: As noted already in Wex et al. (2018), for the relevant pressure and temperature conditions, the presence of kyanite as the result of plagioclase breakdown, to the exclusion of clinozoisite / epidote, implies a water activity of less than ca. 0.004, according to Wayte et al. (1989) (as is also noted again in the current manuscript). The examples from Holsnoy all have extensive development of clinozoisite during eclogite formation.

-

RC: Figure 5: The difference between fracture types I and II is not very clear to me. The magnification at which the image was taken is quite low and therefore it is difficult to see subgrains.

AC: The step-size for this map was 2 micrometres, which is obviously a compromise due to the large area of the garnet, and individual points are still visible in the figure. Unfortunately, we do not have a higher resolution scan for the specific area. We hope that the subgrains are still visible as slight changes in colour and grey-values, as seen and highlighted in the red area. We admit that there is no genetic difference between the proposed fracture sets I and II and have therefore dropped any differentiation between the two.

MC: Figure 5 was changed in regard to the labelling of the fractures and the text was changed accordingly.

---

## Author Response (AR2)

**Editor Comments to the Author:**

Dear Friedrich and coauthors,

Thanks for submitting the revised manuscript, which I will recommend for publication after
a very minor improvement: In the opening sentence of the introduction, you refer to
processes [not being preserved] in the rock record, an important theme. However, your
transition into garnet as a recorder of such processes is somewhat abrupt. Could you pick
this theme up just before the bullet list at the end of the introduction to introduce the list?

Further, lines 232-240 in the revised paper: Please shorten and restrict to what is directly
relevant to your research question.

Thanks a lot,
Florian

**Author response to Editor comments:**

Dear Florian,

Thank you for your recommendation of publication with very minor revisions and for your
suggestions with regard to those minor improvements. We agree that the transition in the
lines following line 82 were too abrupt and have expanded the section immediately before
the bullet points as follows:

Garnets can retain their microstructure and chemical composition during retrograde
deformation and metamorphism and can therefore preserve indicators for seismic events,
which are otherwise possibly erased from the rock record. Here we present a study of
garnet microstructures from lower crustal rocks of the Musgrave Block in Australia, which:
[...]

For the lines 232-240 , we would argue that this description of the observed fracture sets is
already concise (indeed only 8 lines) and crucial to our later interpretation. It is necessary to
establish the existence of the late set II extensional fractures and the latter sentences are
simply a description of the fracture distribution and microstructure seen in Fig 5c.
This description is necessary in the text to highlight to the reader what we believe are the
most important features of the microstructure visible n 5c.

very best regards,
Friedrich and coauthors

We want to thank Matthias Konrad-Schmolke for his critical review and suggestions. Below is a list of
all comments from the reviewer (RC), answers from the authors (AC) and manuscript changes (MC).

**Reviewer 1, general comment**

**RC**: One crucial argument for crystal plasticity in garnet is the observation of dislocation walls that
mark the boundary of one subgrain in the garnet crystals. The authors state that these dislocation
walls are the result of dislocation climb in the crystal (lines 252-253) and therefore indicate the
activity of viscous deformation mechanisms in garnet. I am not entirely convinced that these
dislocation walls are only produced by the migration of dislocations through the crystal, although I
am not aware of studies that demonstrate neither pro nor contra arguments. The fact that the
authors do not cite any references is also not helpful with this regard. However, there is evidence
that such dislocation walls can be generated in undeformed rocks, e.g. during fluid infiltration, such
as demonstrated in Konrad-Schmolke et al., 2018. Of course, fluid infiltration does not play a role in
the rocks presented here, but other mechanisms for the formation of the dislocation walls must be
discussed in this manuscript, as these structures are a fundamental argument for crystal plasticity.
Furthermore, the interpretation of the presence of rotated subgrains in terms of subgrain rotation
recrystallization is, in my opinion, also questionable. Konrad-Schmolke et al., 2007 demonstrate the
presence of subgrains in garnets (and their slight misorientations) in undeformed rocks. In general, I
think that the manuscript would very much benefit from a more indepth discussion of these
features. The papers cited in this review should only serve as examples and I think that there are
many other contributions to the topic that I am not aware of at the moment. However, I think the
manuscript is very well suitable for publication after moderate revisions and a more thorough
discussion.

**AC**: As noted by the reviewer, fluid infiltration does not play a role in the rocks presented here, so
the mechanism proposed in Konrad-Schmolke et al., 2018 cannot be relevant in this case. The rocks
we are considering here are also clearly deformed, with stresses being high enough (at least
transiently) to cause fracturing of garnet. Progressive subgrain rotation by migration of dislocations
into walls bounding such subgrains is a mechanism that has been very widely proposed both in the
material and earth sciences. There is a large body of published work supporting and describing this
mechanism – indeed as noted by the reviewer "I think that there are many other contributions to
the topic". It is not the aim of the current manuscript to provide an exhaustive review be we have
now added the following additional references as a representative selection:

Hobbs, B.E.: Recrystallisation of single crystals of quartz. Tectonophysics, 6, 353-401, 1968.

Passchier, C.W., Trouw, R.A.J.: Microtectonics (2nd Edition), Springer, Heidelberg, 366 pp., 2005.

**Reviewer 1, specific comments**

**RC**: Line 138: if the fractures a dilatant there must be some material in the cracks. Can that be
evaluated?

**AC**: No, these fractures remain empty, as implied by the word "unfilled" in the original text. These
fractures are Mode 1 extensional fractures, which we think open during propagation of the seismic
wave and immediately close, preventing any mineral filling.

**MC:** The word "dilatant" is perhaps better replaced with "extensional", so the text now reads "*An*
*apparent late generation of unfilled extensional fractures […]*". All other similar references to
"dilatant fractures" have also now been changed to "extensional fractures".

**RC**: What about the other, fast diffusing elements, such as Mn and Mg? Differences in diffusion
lengths would indicate different diffusion velocities and thus support the idea of a diffusional
modification.

**AC**: In Figure 4 d) we present the profiles for Fe, Mg, Mn. Fe and Mg show the same diffusion length
as Ca. Mn does not show any measurable modification throughout the crystal.

**MC:** This observation was missing in the text, therefore we added the following sentence for
clarification: "*The length-scale for variation in Fe ($X_{Alm}$) and Mg ($X_{Pyp}$) is identical to that for Ca ($X_{Grs}$),*
*whereas the Mn content ($X_{Sps}$) does not show any variation (Fig 4d)."*

**RC**: Lines 196-197: This diffusional modification is likely due to subgrain boundaries that might or
might not be associated with subgrain rotations. This has been demonstrated in Konrad-Schmolke et
al., 2007 (EJM). This should be discussed or at least mentioned.

**AC**: Since a subgrain is defined by a relative crystallographic rotation (commonly taken arbitrarily as
between ca. 4° and 15°, when it is considered to be a "high-angle boundary" to a "new grain", e.g.
Urai et al., 1986), the generally accepted argument is that subgrain boundaries are always associated
with subgrain rotations, as new dislocations are continuously added to the subgrain boundaries (e.g.
Passchier and Trouw, 2005, p.43). We have added the reference to Konrad-Schmolke et al. (2007), as
well as recent publications of Petley-Ragan et al. (2019), Jamtveit et al. (2018a,b, in press), Engi et al.
(2018), Giuntoli et al. (2018) and Angiboust et al. (2017) when comparing and contrasting our "dry"
results to fracture and diffusion in garnets in deep-seated rocks where fluid infiltration plays an
important role.

We want to thank the anonymous reviewer for his critical review and suggestions. Below is a list of
all comments from the reviewer (RC), answers from the authors (AC) and manuscript changes (MC).

**Reviewer 2, general comment**

**RC**: The main message of this manuscript is the occurrence of crystal plasticity in garnet at
temperatures well below the laboratory derived data for the onset of crystal plastic deformation in
garnet. The authors, therefore, state that laboratory data fail to explain the natural observations. Of
course laboratory experiments are most often very simplified rendering extrapolation to natural
systems rather challenging. Though, I miss a bit the explanation why the laboratory data does not
match natural observations. Is it because the samples in the laboratory were even drier than the
natural rock delaying the onset of crystal plastic deformation in the laboratory? Obviously there
were some fluids present due to the occurrence of biotite. Though in some parts of the manuscript,
the authors state that a Ca-rich garnet forms instead of epidote, because of the low water activity
(line 290). Perhaps there was enough water around to facilitate crystal plastic deformation but not
enough to stabilize epidote? I think it would improve the manuscript to discuss the role of fluids on
crystal plastic deformation in more detail. This might also explain the discrepancy between
laboratory data and the natural observations.

**AC**: We have added text in several places to expand the discussion of the apparent discrepancy with
laboratory data, in particular considering the potential effects of strain rate and role of fluids. We
agree that this should have been treated in more detail, which is why we now have a rather more
nuanced approach, considering factors that may have an influence rather than just stating that there
is a difference.

**RC:** Line 71: In this context I think crystal plastic deformation instead of ductile deformation is more
appropriate.

**AC**: agree

**MC:** changed sentence: *"between brittle and crystal plastic deformation of garnet"*

**RC:** Lines 276-278: Did you investigate/find garnet crystals that were cut by a pseudotachylyte? Both
studies that you cite, Austrheim et al. (2017) and Papa et al. (2018), demonstrate garnet crystals that
are situated right next to a pseudotachylyte-bearing fault. I mention this, because as strain rate and
stresses decrease very rapidly with increasing distance, the required stresses and/or strain rates at a
few mm to the fault might not be sufficient anymore to extensively fragment garnet.

**AC**: In the text, we clearly state that *"Granulite facies garnet porphyroclasts in Musgravian
peraluminous gneisses mylonitized during the Petermann Orogeny are almost invariably fractured,
irrespective of their proximity to pseudotachylyte (Fig. 3)."* This is different than what was observed
in the examples of Austrheim et al. (2017) and Papa et al. (2018) mentioned above, which is why we
made such a clear statement originally. On the basis of this observation, we argue in the text that
the whole rock was affected by high stresses during transient seismic events and that garnet
fracturing is not restricted to the localized damage zone of a propagating fracture (Petley-Ragan et
al, 2019; Austrheim et al., 2017) or thermal shock immediately adjacent to the high temperature
pseudotachylyte (Papa et al., 2018).

**RC:** Line 286: Delete 'of some'.

**AC**: agree

**MC:** Typing error corrected.

**RC:** Lines 291-292: So everything is dry, but suddenly there is biotite? You should discuss the
presence/absence of hydrous minerals a bit more.

**AC**: Biotite is a typical mineral of granulite facies assemblages up to the point of melting (with biotite
then providing the water for "anhydrous" melting) and even then biotite is a common mineral in the
restite assemblage. "Kinzigite", which is a typical "dry" lower-crustal granulite facies rock, is actually
defined as having garnet + biotite. As noted by Pennacchioni and Cesare (1997), under upper
amphibolite facies conditions, newly grown biotite can actually act as a sink for any free water
available and the same will be true for the "dry" high pressure upper amphibolite ("sub-eclogitic")
facies conditions relevant to the current study.

Pennacchioni, G. & Cesare, B., 1997. Ductile-brittle transition in pre-Alpine amphibolite facies
mylonites during evolution from water-present to water-deficient conditions (Mont Mary nappe,
Italian Western Alps). Jour. Metm. Geol. 15, 777-791.

**RC:** Lines 304-305: Shimada et al. (1983) experimentally investigated that the angle changes from
around 30 to approx. 45° with increasing pressure.

**MC:** The reference was added to the text: *"This plot is only qualitative, since the angle of internal*
*friction could decrease towards higher pressure (Shimada et al., 1983)."*

**RC:** Lines 311-313: See comment above. As water seems important you should perhaps quantify the
amount of water? There is some water present in the other field studies mentioned, but not very
much. How should the presence of a fluid help to fragment the rock?

**AC**: As noted already in Wex et al. (2018), for the relevant pressure and temperature conditions, the
presence of kyanite as the result of plagioclase breakdown, to the exclusion of clinozoisite / epidote,
implies a water activity of less than ca. 0.004, according to Wayte et al. (1989) (as is also noted again
in the current manuscript). The examples from Holsnoy all have extensive development of
clinozoisite during eclogite formation.

**RC:** Figure 5: The difference between fracture types I and II is not very clear to me. The magnification
at which the image was taken is quite low and therefore it is difficult to see subgrains.

**AC**: The step-size for this map was 2 micrometres, which is obviously a compromise due to the large
area of the garnet, and individual points are still visible in the figure. Unfortunately, we do not have
a higher resolution scan for the specific area. We hope that the subgrains are still visible as slight
changes in colour and grey-values, as seen and highlighted in the red area. We admit that there is no
genetic difference between the proposed fracture sets I and II and have therefore dropped any
differentiation between the two.

**MC:** Figure 5 was changed in regard to the labelling of the fractures and the text was changed
accordingly.

**MANUSCRIPT INCLUDING CHANGES**

[revised manuscript text omitted]